# Safety of Dual Orexin Receptor Antagonist Daridorexant: A Disproportionality Analysis of Publicly Available FAERS Data

**DOI:** 10.3390/ph17030342

**Published:** 2024-03-06

**Authors:** Giuseppe Cicala, Maria Antonietta Barbieri, Giulia Russo, Francesco Salvo, Edoardo Spina

**Affiliations:** 1Department of Clinical and Experimental Medicine, University of Messina, 98125 Messina, Italy; gcicala@unime.it (G.C.); giuliarusso.ab@gmail.com (G.R.); espina@unime.it (E.S.); 2Regional Pharmacovigilance Centre of Bordeaux, Department of Medical Pharmacology, Centre Hospitalier Universitaire de Bordeaux, F-33000 Bordeaux, France; francesco.salvo@u-bordeaux.fr; 3AHeaD Team, U1219, Bordeaux Population Health, National Institute for Health and Medical Research (Institut National de la Santé et de la Recherche Médicale, INSERM), Université de Bordeaux, F-33000 Bordeaux, France

**Keywords:** daridorexant, insomnia, orexin, pharmacovigilance databases

## Abstract

Daridorexant (dari), as the first dual orexin receptor antagonist (DORA) marketed in Europe, offers a novel therapeutic approach to insomnia. However, data regarding its real-world safety are scarce. Thus, this study was aimed at assessing its safety profile using a large-scale pharmacovigilance database. Dari-related adverse drug reaction (ADR) reports from the Food and Drug Administration Adverse Event Reporting System were scrutinized, and ADRs were selected using reporting odds ratio (ROR) as a measure of disproportionality. Frequencies of events related to dari were compared to all other drugs (reference group, RG1) and only to other DORAs (RG2). Only significant disproportionalities to both RGs were evaluated in-depth. A total of 845 dari-related reports were selected; nightmares (*n* = 146; dari vs. RG1: ROR = 113.74; 95%CI [95.13, 136]; dari vs. RG2: ROR = 2.35; 95 CI% [1.93, 2.85]), depression (*n* = 22; dari vs. RG1: 2.13; [1.39, 3.25]; dari vs. RG2: ROR = 2.31; 95 CI% [1.45, 3.67]), and hangover (*n* = 20; dari vs. RG1: ROR = 127.92; 95 CI% [81.98, 199.62]; and dari vs. RG2: 3.38; [2.04, 5.61]) were considered as safety signals. These data provide valuable insights into the real-world safety profile of daridorexant, supporting the existence of safety signals related to nightmares, depression, and hangovers.

## 1. Introduction

The Fifth Edition of the Diagnostic and Statistical Manual of Mental Disorders defines insomnia as a condition characterized by a predominant dissatisfaction with sleep quantity or quality [1]. In terms of primary symptoms, this condition can be characterized by difficulties initiating or maintaining sleep, as well as by early morning awakenings with an inability to return to sleep. These symptoms must be present for at least three nights per week and persist for a period of at least three months despite the presence of adequate opportunities to sleep to meet the diagnostic criteria for insomnia disorder (ID) [1]. ID appears to be among the most widespread and enduring medical conditions, as it is estimated to afflict approximately 10% of the global demographic [2]. An important consequence of ID is the impairment of daytime functioning [3]. It, in fact, could substantially compromise various aspects of a patient’s life, including social interactions, occupational performance, educational achievement, and behavioral outcomes [4]. In addition to that, ID was also correlated with an increased risk of chronic noncommunicable diseases such as hypertension, diabetes, and depression [5]. Current treatment guidelines for ID recommend, as a first-line option, the so-called cognitive behavioral therapy for insomnia (CBT-I) [6,7]. However, maintaining adherence to this type of intervention can be challenging. Thus, it is frequently necessary to support patients with pharmacological interventions [8].

Currently, available treatment options for treating ID include benzodiazepines (BDZs), agonists of the benzodiazepine receptor (Z-drugs), some antidepressants (i.e., mirtazapine and trazodone), and melatonin agonists [9]. However, BDZs, Z-drugs, and other hypnotic medications are acknowledged for their propensity to induce physical dependency in patients undergoing extended treatments. In addition to that, therapy discontinuation can often lead to protracted withdrawal symptoms that persist for several months [10,11]. The dual orexin receptor antagonists (DORAs) represent an innovative treatment strategy that could help to treat ID symptoms while reducing the burden of tolerability problems [12]. Orexins are two neuropeptides (orexin A and orexin B) secreted from neurons originating from the lateral hypothalamus that project to various brain regions. Orexin neurotransmission is mediated through two types of postsynaptic G-protein-coupled receptors: orexin 1 (Ox1R) and orexin 2 (Ox2R). Orexins play a key role in the regulation of numerous physiological functions, including arousal, cognition, stress responses, appetite regulation, and metabolism [13]. DORAs act by exercising a blockade of both types of Ox receptors, reducing the intensity of wakefulness stimuli and promoting sleep onset [12,14]. This mechanism of action prevents the broader inhibition of neuronal pathways and should help to reduce associated side effects (such as next-morning residual sleepiness, falls, motor incoordination, tolerance, and physical dependence) that are inherent to GABA-A modulators such as BDZs and Z-Drugs. Currently, there are three market-approved DORAs, suvorexant, lemborexant, and daridorexant (dari), that primarily differ in their pharmacokinetic (PK) properties [15].

Dari is the latest DORA introduced on the market and the only one to be authorized for the treatment of insomnia by the European Medicines Agency (EMA) [16]. It is characterized by a pharmacokinetic profile presenting fast absorption and a shorter half-life (8 h) compared to the other DORAs. This should enable further reduction in next-morning sleepiness while maintaining the same level of control over nighttime sleep [17]. In its premarketing phase, dari showed an overall encouraging safety profile, with the most frequently observed adverse drug reactions (ADRs) being relative to headache, somnolence, and dizziness [18]. However, some concerns regarding tolerability aspects that are challenging to evaluate in randomized controlled trials, such as the increased frequency of suicidal ideation-related ADRs, are currently being investigated [18,19,20,21]. Pharmacovigilance evaluations based on disproportionality analyses have proven to be valid tools for providing insightful safety data regarding newly marketed drugs or formulations [22,23]. However, to our knowledge, studies conducted using spontaneous reporting data regarding the safety of dari, especially those comparing it to the other DORAs, are still limited. Considering this, our aim was to investigate the overall safety profile of this new therapeutic option. This evaluation was realized using individual case safety reports (ICSRs) collected in the US Food and Drug Administration (FDA) Adverse Event Reporting System (FAERS).

## 2. Results

The initial FAERS dataset, covering from the database inception to 31 March 2023, contained 19,514,139 records. After the removal of obsolete versions of ICSRs, duplicates, and undescriptive (i.e., presenting only generic terms such as “unevaluable event” or “no adverse drug reaction”) ICSRs, 15,676,304 ICSRs were left. Among those, the ones in which dari was considered as a primary suspect drug were 845, while 8986 had other DORAs as a suspected drug (Figure 1).

### 2.1. Descriptive Analysis

Most dari-related ICSRs were found to be issued by consumers (*n* = 712; 84.3%). When comparing these ICSRs to those relative to the other DORAs, a significant difference in the proportion of consumer-reported ICSRs (*n* = 712; 84.6% vs. *n* = 3426; 38.1%; *p* < 0.001) over those reported by healthcare professionals (*n* = 131; 15.5% vs. *n* = 5549; 61.8%; *p* < 0.001) was observed.

Regarding demographical characteristics, dari-related ICSRs showed a woman-to-man patient ratio of 1.9 (*n* = 527; 62.4% vs. *n* = 275; 32.5%). Patient age data were available in 42.3% of the examined ICSRs. Our analysis showed that ICSRs relative to dari had a lower median age value when compared to those relative to the other DORAs (60, interquartile range, IQR [51–60] years vs. 64 [50–74] years; *p* = 0.003). Furthermore, a significantly higher frequency of patients in the 50 to 64 years age group was observed in dari-related ICSRs (*n* = 101; 38.7% of ICSRs with age information vs. *n* = 1065; 27.3% of ICSRs with age information; *p* < 0.001). As far as the reported ADR characteristics were concerned, most dari-related ICSRs were described as not serious ADRs (*n* = 805; 95.3%). The percentage of serious ADR cases in dari-related ICSRs was lower than the one observed in other DORA-related ICSRs (*n* = 40; 4.7% vs. *n* = 1559; 17.4%; *p* < 0.001). Among serious ADRs, those categorized as “other serious important medical events” (*n* = 26; 3.1%) were the most represented, followed by ICSRs in which the death of the patient was reported (*n* = 9; 1.1%) and ICSRs that described an ADR causing or prolonging a hospitalization (*n* = 3; 0.4%). See Table 1 for further details.

In terms of ADR categories at the Medical Dictionary for Regulatory Activities (MedDRA^®^) High Level Group Terms (HLGTs), those observed with the higher frequency were sleep disorders and disturbances (*n* = 326, 38%) with the most frequently associated preferred terms (PTs) being nightmare (*n* = 146, 17.3%), insomnia (*n* = 80; 9.5%), abnormal dreams (*n* = 64; 7.6%), somnambulism (*n* = 18, 2.1%), and middle insomnia (*n* = 15; 1.8%). These ADRs were followed by those relative to therapeutic and nontherapeutic effects (*n* = 311; 36.8%), namely drug ineffective (*n* = 233; 27.6%), therapeutic product effect incomplete (*n* = 33; 3.9%), therapeutic product effect delayed (*n* = 18; 2.1%), therapeutic product effect variable (*n* = 8; 0.9%), and therapeutic product effect decreased (*n* = 8; 0.9%). The third class of frequently observed ADRs were those relative to general system disorders (*n* = 194; 23%), specifically feeling abnormal (*n* = 64; 7.6%), fatigue (*n* = 45; 5.3%), hangover (*n* = 20; 2.4%), illness (*n* = 17; 2%), and malaise (*n* = 12; 1.4%). Full details regarding the observed HLGT distribution for dari-related ICSRs are available in Appendix A. Data concerning the full distribution of ADRs at the PT level for each HLGT are available in Appendix A.

### 2.2. Disproportionality Analyses

#### 2.2.1. MedDRA^®^ High Level Group Term Analysis

The reporting odds ratio (ROR) with a 95% confidence interval (CI) was used as a disproportionality measure to select ADRs of potential interest.

ADRs were first considered at MedDRA^®^ HLGTs. HLGTs in dari-related ICSRs were compared to both ICSRs relative to all other drugs (Reference Group 1; RG1) or to those related to another DORA (suvorexant or lemborexant) as a suspected drug (Reference Group 2; RG2). HLGTs related to significant ROR for both RGs were further investigated.

Significantly higher odds of reporting were observed when in relation to the HLGTs: sleep disorders and disturbances (*n* = 326; dari vs. RG1: ROR = 26.42; 95%CI [23.01, 30.35]; dari vs. RG2: 1.84; [1.59, 2.13]), product quality, supply, distribution, manufacturing, and quality system issues (*n* = 69; dari vs. RG1: 3.98; [3.11, 5.08]; dari vs. RG2: 4.12; [3.1, 5.47]) and salivary gland conditions (*n* = 14 dari vs. RG1: 3.38; [1.99, 5.73]; and dari vs. RG2: 2.21; [1.24, 3.94]). See Table 2 for further details.

#### 2.2.2. MedDRA^®^ Preferred Term Disproportionality Analysis and Case-by-Case Assessment

Disproportional HLGTs were then examined at a more detailed level, the MedDRA^®^ PT one (e.g., the headache HLGT contained the headache PT as well as migraine). Additionally, their expectedness was evaluated using the US FDA or European Summary of Product Characteristics (SmPCs). Data on co-reported PTs, dechallenge, rechallenge, and time-to-onset (TTO) for each double disproportionality signal were obtained using a case-by-case assessment methodology.

Disproportionally reported PTs in the HLGT sleep disorders and disturbances included the following: nightmare (*n* = 146 dari vs. RG1: ROR 113.74; 95%CI [95.13, 136]; dari vs. RG2: 2.35; [1.93, 2.85]), insomnia (*n* = 80; dari vs. RG1: 7.41; [5.88, 9.33]; dari vs. RG2: 1.94; [1.51, 2.49], and hypnagogic hallucination (*n* = 3 dari vs. RG1: 267.22; [85.32, 836.95]; dari vs. RG2: 4; [1.06, 15.1]). See Table 3 for further details.

The case-by-case assessment for these ADRs showed, as frequently, co-reported PTs with nightmare: drug ineffective (*n* = 32; 21.9%), hallucination (*n* = 10; 6.8%), and hangover (*n* = 8; 5.5%). A positive dechallenge was observed in 20.5% of nightmare ICSRs (*n* = 30), and a negative dechallenge in 3.4% (*n* = 5). TTO data were available in 20 nightmare ICSRs (13.7%) and showed a median of 1 IQR [1–1] day. Considering insomnia, frequently co-reported PTs included the following: product availability issue (*n* = 46; 57.5%), drug ineffective (*n* = 7; 8.8%), and feeling abnormal (*n* = 7; 8.8%). A positive dechallenge was reported in 20.5% of insomnia ICSRs (*n* = 30), and a negative dechallenge in 3.4% (*n* = 5). TTO data were present in six ICSRs (7.5%) with an observed median of 1 IQR [1,2,3,4,5,6,7,8,9,10,11,12,13] day. As far as hypnagogic hallucination was concerned, co-reported PTs included confusional state, auditory hallucination, and nightmare (all with *n* = 1; 33.3%). A positive dechallenge was observed in two cases (66.7%), while no rechallenge and TTO data were available for these ICSRs.

In the HLGT salivary gland conditions, the only observed PT was dry mouth (*n* = 14 dari vs. RG1: ROR = 4.17; 95%CI [2.46, 7.07]; dari vs. RG2: 2.31; [1.29, 4.14]. This ADR is not currently reported neither in the European or FDA SmPCs. The case-by-case assessment showed that frequently co-reported PTs included the following: drug ineffective (*n* = 5; 35.7%), nocturia (*n* = 2; 14.3%), and nightmare (*n* = 2; 14.3%). A positive dechallenge was reported in six ICSRs (42.8%). No TTO data were available.

As far as the HLGT product quality, supply, distribution, manufacturing, and quality system issues was concerned, the disproportional PTs were as follows: product packaging difficult to open (*n* = 7; dari vs. RG1: ROR = 104.91 95%CI [49.76, 221.20] dari vs. RG2: 5.35; [2.15, 13.3]), and product availability issue (*n* = 51; dari vs. RG1 97.47; [73.39, 129.45] dari vs. RG2: 32; [18.61, 55.04]). See Appendix A for more information. Neither ADRs were reported in either the dari European or FDA SmPCs. The case-by-case assessment data showed that frequently co-reported PTs with product packaging difficult to open included drug ineffective (*n* = 3; 42.9%), headache (*n* = 2; 28.6%), and yawning (*n* = 1; 14.3%). PTs frequently co-reported with product availability issue were insomnia (*n* = 46; 90.2%), drug ineffective (*n* = 5; 9.8%), and nightmare (*n* = 3; 58.8%).

The disproportional PTs reported in the HLGT headaches were headache (*n* = 59; dari vs. RG1: ROR = 2.27; 95%CI [1.74, 2.95]; dari vs. RG2: 1.52; [1.15, 2.01]) and migraine (*n* = 8; dari vs. RG1: 2.05; [1.02, 4.12]; dari vs. RG2: 2.25; [1.05, 4.84]). See Appendix A for more information. These ADRs are already listed in European or FDA SmPCs. Frequently co-reported PTs that emerged from the case-by-case assessment of headache ICSRs included the following: drug ineffective (*n* = 16; 27.1%), somnolence (*n* = 9; 15.2%), and fatigue (*n* = 8; 13.6%). A positive dechallenge was observed in 49.2% of these cases (*n* = 29) and a negative one in 3.4% (*n* = 2). TTO data were available in 17 ICSRs (28.8%) with a median observed value of 1 IQR [1–1] day. Considering migraine, frequently co-reported PTs included the following: tremor, somnolence, and product prescribing error (all with *n* = 1; 12.5%). A positive dechallenge was observed in 37.5% of cases (*n* = 3). TTO data were available in only two reports, both showing an interval of 2 days between treatment initiation and ADR onset. No rechallenge data were available for both headache and migraine.

As far as PTs belonging to the HLGT depressed mood disorders and disturbances, those were as follows: depression (*n* = 22; dari vs. RG1: ROR = 2.13; 95%CI [1.39, 3.25] and depressed mood (*n* = 3; dari vs. RG1: 1.36; [0.44, 4.24]); see Appendix A for more information. Both ADRs are acknowledged in European or FDA SmPC. Frequently co-reported PTs with depression were drug ineffective (*n* = 7; 31.8%), suicidal ideation (*n* = 5; 22.7%), and insomnia (*n* = 4; 18.2%). A positive dechallenge was observed in 45.5% of depression cases (*n* = 10) and a negative one in 9.1% (*n* = 2). No rechallenge data were available. TTO data were available in six ICSRs (27.3%), with a median value of 5 days IQR [1,2,3,4,5,6,7,8,9,10,11].

Considering the HLGT General system disorders NEC, disproportional PTs were feeling abnormal (*n* = 64 dari vs. RG1: ROR = 6.19; 95%CI [4.8, 7.99]), followed by fatigue (*n*= 45 dari vs. RG1: 1.41; [1.05, 1.91]), hangover (*n* = 20; dari vs. RG1: 127.92; [81.98, 199.62]), and energy increased (*n* = 5; dari vs. RG1: 18.7; [7.76, 45.06]); see Table 4 for further details.

The case-by-case assessment for feeling abnormal ICSRs highlighted frequently co-reported PTs such as drug ineffective (*n* = 17; 26.6%), fatigue (*n* = 14; 21.9%), and headache (*n* = 8; 12.5%). These ICSRs were characterized by a positive dechallenge in 25 cases (39.1%) and a negative dechallenge in 2 cases (3.1%). TTO data were available in 16 of these ICSRs (25%) with an observed median value of 1 IQR [1–1] day. Considering fatigue frequently co-reported PTs included somnolence (*n* = 22; 48.9%), drug ineffective (*n* = 18; 40%), and feeling abnormal (*n* = 14; 31.1%). A positive dechallenge was observed in 57.8% of fatigue-related ICSRs (*n* = 26). In terms of rechallenge, one ICSR (2.2%) showed a positive rechallenge and another ICSR a negative one. Twelve fatigue-related ICSRs (26.7%) presented TTO data with a median value of 1 IQR [1–1] day. Focusing on hangover, the frequently co-reported PTs were nightmare (*n* = 8; 40%), drug ineffective (*n* = 5; 25%), and somnolence (*n* = 3; 15%). A positive dechallenge was observed in 20% of hangover ICSRs (*n* = 4). TTO data were available in only two hangover-related ICSRs with intervals of 1 or 2 days. The evaluation of energy increase-related ICSRs showed frequently co-reported PTs insomnia, drug ineffective (both with *n* = 2; 40%), and therapeutic response unexpected (*n* = 1; 20%). A positive dechallenge was observed in three cases (60%), while TTO data were available in two cases with a 1 day onset time.

Full data regarding disproportionality evaluations of observed PTs for each HLGT are available in Appendix A.

## 3. Discussion

This study analyzed the currently available real-world safety data for dari reported in the FAERS database and compared them to the ones relative to the other DORAs. This new targeted drug class represents an important step forward in ID pharmacological treatment. DORAs promise, in fact, to offer a therapeutic alternative that could improve critical tolerability aspects of commonly used medications in this setting. Furthermore, dari, being the only DORA authorized by both the FDA and EMA, could represent a new therapeutic alternative for a portion of patients who previously did not have access to these treatments. Our analysis points out increased odds of reporting for potentially important ADRs such as nightmares and worsening of depression. Considering this and given the current lack of real-world safety studies regarding this drug, the result of our analysis could represent a useful tool to help clinicians.

### 3.1. Descriptive Analysis

A significantly higher frequency of ICSRs reported by patients was observed when comparing dari-related ICSRs to those relative to the other DORAs. This can be viewed as a consequence of the high level of attention that dari received in its first year of market presence. Furthermore, dari is currently being actively promoted, especially in the US, which could incentivize consumer-driven reporting. Multiple studies have highlighted how the media could strongly influence ADR reporting by consumers [24].

A significantly higher proportion of dari-related ICSRs involved patients in the 50 to 64 years age group compared to those relative to the other DORAs. These data are in line with the common clinical practice in which consolidated therapeutic options are always preferred for the treatment of older, thus more fragile, patients [24,25].

A significantly lower frequency of serious ADRs was observed when comparing dari-related ICSRs to those relative to the other DORAs. This data could be intended as confirmation of premarketing studies depicting an overall good safety profile. However, the shorter market permanence of dari should be taken into account. Indeed, as the market permanence of a drug increases, the propensity for reporting nonserious ADRs diminishes due to their increasing recognition and familiarity among prescribers [25]. Therefore, the ratio between ADRs classified as serious and those classified as nonserious can undergo significant variations. As far as the seriousness characteristics of ADRs are concerned, even if differences were observed compared to the other DORAs, the extremely limited number of ICSRs presenting serious ADRs makes an unbiased evaluation of these data difficult.

The relatively high frequency of observation for terms regarding drug ineffectiveness represents an interesting occurrence. Several reasons, such as patient skepticism and initial administration errors, could contribute to reporting this kind of ADR. Furthermore, literature sources highlight that patient-reported ICSRs, which represent the majority of dari-related ICSRs, have a higher ratio of drug ineffectiveness when compared to those reported by healthcare professionals [26,27]. These considerations can also be applied to the high number of ICSRs reporting insomnia observed, pointing towards the existence of an indication bias. The previously discussed ADRs are, thus, more likely to be attributed to ineffective control of preexisting ID conditions rather than being a direct result of drug administration.

### 3.2. Disproportional ADRs

When compared to the rest of the FAERS database, disproportionalities concerned mostly ADRs already known for other DORAs. This data depicts a globally comparable safety profile for all DORAs. However, some differences seem to be present and merit further attention.

#### 3.2.1. Hangover Manifestations

Hangover and fatigue were found to be disproportionally reported in dari-related ICSRs when compared to both RGs. This data should be viewed in light of some considerations. Dari was selected among several drug candidates thanks to an expected effect duration of around 8 h with a dose of 25 mg. This choice was made in order to minimize “next morning” residual effects [28]. However, some people may still experience “next morning” residual effects, such as somnolence, when treated with dari [18,29]. In addition to that, in one-fourth of ICSRs, hangover also reported ineffectiveness or effectiveness reduction. Moreover, because dari was designed to minimize such ADRs, their occurrence is more likely to garner increased attention than other DORAs. Considering this, we cannot rule out the presence of a notoriety bias regarding this ADR.

#### 3.2.2. Dry Mouth

Dry mouth was also disproportional for dari compared to both RGs. Currently, this ADR is not reported either in the European or in the US FDA SmPCs. Only the FDA documentation relative to suvorexant reported dry mouth as an ADR observed in premarketing studies [30]. Although apparently mild, the impact of this ADR on patients’ quality of life has to be carefully considered. Indeed, literature sources point out that xerostomia or self-reported sensations of dry mouth can be correlated with insufficient quality of sleep [31,32]. Furthermore, from a case-by-case analysis, we were able to observe that 35.7% of ICSRs presenting dry mouth as an ADR also highlighted the ineffectiveness of the drug. Thus, the reporting by patients of a feeling of dry mouth should be, in our opinion, carefully considered by the clinicians as it might be related to a sub-optimal control of the quality of sleep.

#### 3.2.3. Nightmares

Nightmares were found to be disproportional for dari both when considering the rest of the database and to ICSRs relative to the other DORAs. The possibility of the onset of this type of ADR is currently not reported in the EMA SmPC of dari but just in the FDA one. Furthermore, data associating these ADRs to other DORAs is limited to the results of two clinical trials conducted for lemborexant [33]. The onset of nightmares was deemed as an uncommon occurrence in both studies. Indeed, in the SUNRISE-1 trial, only 0.6% (3 out of 534) of patients treated with lemborexant developed nightmares [34]. While considering the SUNRISE-2 trial, 1.8% of patients treated with lemborexant (11 out of 628) presented nightmares [35]. These ADRs should not be underestimated as the onset of nightmares in patients with sleep disorders has been linked to increased distress levels and, in some cases, to the development of secondary forms of ID [36]. The mechanism behind the onset of nightmares might be linked to the actions that DORAs as a class have on patients’ global sleep architecture. Indeed, one of the supposed effects of DORAs is promoting the REM phase of sleep [37]. This increase in REM sleep time could facilitate the recalling of dream content and its remembering by the patient [38]. Furthermore, other ADRs, like headaches or migraines, are possible risk factors for nightmares [39]. However, after an in-depth evaluation of nightmare cases (*n* = 146), we observed that 21.9% of those also reported an ineffectiveness of the drug. Thus, the impact of inadequate control or worsening preexisting conditions cannot be excluded. In addition to that, we observed that available TTO data (13.7%) indicated short onset median time (1 day). These data, even if limited, surely remark on the need to monitor the initial stages of ID pharmacological treatments closely. Furthermore, a need to further investigate these issues with focused studies emerges from our analysis. An initial step in this direction should involve epidemiological studies based on large-scale databases. These studies might be particularly fitting considering the low incidence rates noted in clinical trials for lemborexant and the limited availability of real-world data. Once a deeper understanding of the problem is achieved, the next step should involve short- and long-term prospective studies to obtain first-hand clinical practice data.

#### 3.2.4. Worsening of Depression

Depression was found to be disproportional for dari both when considering the rest of the database and to ICSRs relative to the other DORAs. The risk of worsening pre-existing depression condition might represent an important safety aspect of dari. Previously, another DORA, suvorexant, has been associated with seemingly dose-dependent worsening depression and suicidal ideation [40]. Additionally, general warnings regarding these ADRs are present in the SmPCs [18,29]. Furthermore, previous studies showed that suicidal patients affected by major depressive disorder (MDD) present lower orexin levels than those without a diagnosis of MDD [41]. However, the results of a recently published real-world study seem to underline a certain coherency between DORAs and Z-drugs in terms of suicidal ideation ADRs [21]. Furthermore, sleep disorders are renowned among the core symptoms of depression [42]. Thus, the possibility of the depression worsening as a consequence of the insufficient control of these disturbances cannot be ruled out. As far as suicidal ideation is concerned, the intricate nature of these clinical presentations makes drawing unbiased conclusions a task beyond the capability of a disproportionality analysis-based study. In this context, the employment of techniques like network analyses could aid in the assessment of these manifestations. Indeed, the potential to provide quantifiable data on relationships among ADRs, which these techniques seem to possess, could allow the achievement of a more holistic comprehension of these phenomena.

#### 3.2.5. Other ADRs of Interest

An almost significant ROR comparing dari to RG1 was observed for the HLGT “allergic conditions”. Hypersensitivity was the only disproportional event (Appendix A). In three cases, this was co-reported with rash, while in the other two cases, it was reported with pruritus and erythema. This common pattern suggests further investigations to properly assess the role of dari in allergic or cutaneous reactions and their clinical impact.

### 3.3. Strengths and Limitations

The study is based on real-world data collected from the FAERS database, providing a practical insight into the safety of dari. The fact that dari is the only drug in its class to be authorized by the FDA and EMA makes it particularly interesting to healthcare professionals. To the best of our knowledge, this is the first study evaluating real-world safety data for this drug in comparison to those relative to the other DORAs using a large-scale pharmacovigilance database. Although our findings provide a comprehensive perspective in the evaluation of dari-related ADRs, the results of the present study should be interpreted in the light of some limitations.

FAERS data are based on spontaneous ICSRs; thus, the presence of duplicate ICSRs, reporting errors, and high variability in the general quality and completeness of reported data should be considered. We acknowledge these factors, and we adopted a cautious approach in ICSRs where data were insufficient, often resulting in the exclusion of cases. Other FAERS-specific limitations regarded the TTO, dechallenge, and rechallenge data. These data were present in a limited number of reports and provided on a “drug-event” basis rather than a “drug-single ADR” basis. However, the type of ADRs examined usually present themselves without a large time between the various symptoms. Thus, the impact of these limitations on our analysis should be limited.

Disproportionality analyses are commonly affected by biases such as the notoriety bias and confounding by indication [43]. As far as notoriety bias is concerned, its impact on our analysis cannot be underestimated. We acknowledge this, and in order to minimize this influence, we adopted an extremely cautious approach in the evaluation of ADRs more prone to this phenomenon. Concerning the confounding by indication bias, the implementation of two RGs in our study, the second of which constituted drugs with the same mechanism of action of dari and approved for the same indication, should allow for the mitigation of this issue. However, in this case, given the complexity of the conditions underlying ID, caution was applied in the interpretation of the results.

Furthermore, it is crucial to approach pharmacovigilance data with consideration of certain technical issues, including under-reporting compared to the global clinical population and the challenge of identifying confounding factors. This suggests that the reported ADRs may only represent a partial, and possibly underrepresented, percentage of all ADRs occurring in routine clinical practice. Additionally, the absence of data on the number of patients effectively treated with these drugs during the specified period (i.e., the denominator for calculating incidence fractions) precludes the calculation of incidence rates.

## 4. Materials and Methods

### 4.1. Data Source and Extraction Criteria

A retrospective observational study was conducted by using ICSRs collected in the FAERS database. FAERS is a widely utilized publicly accessible database comprising over 19 million ICSRs from patients, healthcare professionals, and pharmaceutical companies across the United States, Europe, and Asia. These ICSRs, aside from unique identification numbers, can include information, dates of ICSR submission and ADR occurrences, reporting country, primary source qualifications, patient characteristics (such as gender, age, and weight), suspected and concurrent medications along with their prescribed uses, ADR descriptions, and their seriousness [44,45].

Data relative to ICSRs were extracted from the ASCII files made publicly available each quarter by the FDA (https://fis.fda.gov/extensions/FPD-QDE-FAERS/FPD-QDE-FAERS.html, accessed on 15 May 2023). Files corresponding to a time interval ranging from Q1 2004 to Q1 2023 have been downloaded and preprocessed. In case of the presence of multiple versions of a single ICSR, only the latest one was considered for the analyses. Duplicate ICSRs were recognized through overlapping data in crucial fields such as event dates, gender, age, reporting country, body weight, reported ADRs, and suspected active substances. In addition to that, ICSRs linked to literature sources were also excluded. Furthermore, ICSRs not providing a description of an ADR (i.e., presenting only terms such as unevaluable event or no adverse event) were excluded.

ICSRs presenting a unique case identifier code and having a primary suspect drug dari were extracted and considered for the analyses. In addition to those uniquely identifiable ICSRs presenting as primary suspect drugs, the other currently approved DORAs (suvorexant and lemborexant) were also extracted to compare their characteristics with the dari-related ones. No restrictions based on gender or age group reported in the ICSRs were applied for the extraction.

All the observed ADRs have been categorized in accordance with the MedDRA^®^. MedDRA^®^ provides a multiaxial hierarchical classification articulated in five levels. The terms range from the more specific Lowest Level Term to the more encompassing System Organ Class. For this analysis, ADRs were at first regrouped in HLGTs and then examined at the PT level [46]. MedDRA^®^ is developed under the supervision of the International Council for Harmonisation of Technical Requirements for Pharmaceuticals for Human Use.

### 4.2. Data Analyses

#### 4.2.1. Descriptive Analysis

A descriptive statistical analysis was carried out to examine the demographic and clinical attributes of FAERS ICSRs related to dari in comparison to those related to other DORAs. This analysis encompassed patient characteristics (such as gender and age), primary sources of information, year of reporting, reporting countries, as well as ADR characteristics, including seriousness. Continuous variables were presented as medians (IQR), while categorical variables were expressed as absolute values (percentages). Differences in frequencies of the categorical variables were assessed using Pearson’s chi-square test on a 2 × 2 contingency table with Yates’ continuity correction. Continuous variables were evaluated using the Mann–Whitney U test. A statistically significant threshold was set at a *p*-value of <0.001 for all analyses.

#### 4.2.2. Disproportionality Analysis

The ROR with the corresponding 95%CI was used as a disproportionality measure. The statistical significance threshold was set at values > 1 of the lower ROR 95% CI limit, with a minimum of three ICSRs for each drug–event pair.

To evaluate significant disproportionalities in the odds of ADR, reporting a two-step process was implemented: At first, a disproportionality analysis was conducted with ADRs regrouped at the HLGT. Dari-related ICSRs were compared to the rest of the database (RG1). In addition, dari-related ICSRs were compared to RG2 constituted by ICSRs presenting suvorexant and lemborexant as suspected drugs. Disproportionally reported HLGTs were further investigated only if significant results emerged from the comparison with both the reference groups. As a second step, for each disproportional HLGT, disproportional ADRs were investigated at the PT Level. As for the previous step, disproportional ADRs were further analyzed only if significant differences were observed in comparison to both reference groups.

#### 4.2.3. Case-by-Case Assessments, TTO Calculation, and ADR Expectedness Evaluation

Qualitative case-by-case assessments were conducted to further characterize the observed double disproportionality signals. These evaluations were performed by examining ADRs reported concomitantly to the disproportional ones. The presence of dechallenge and rechallenge data in ICSRs presenting disproportional ADRs to both RGs was also evaluated in this context. In addition to that, the TTO of the ADR cases was calculated as the difference in days between the event date and the starting date of the therapy [47]. The expectedness evaluation of the observed ADRs for dari was performed on the basis of the data contained in the European or US FDA SmPCs.

All statistical analyses were performed using SPSS version 28. The present study is reported in accordance with the “Reporting of A Disproportionality analysis for drUg Safety” (READ-US) statement [48].

## 5. Conclusions

This study analyzed real-world safety data for dari as reported in the FAERS database, with a particular focus on comparing it to other drugs of the same class. Dari, as the only drug of its class to be authorized by both the FDA and EMA, could represent a new treatment option for many patients. Our data seem to point towards a safety profile overall in line with literature data and previous studies on other DORAs. However, the possible presence of still not entirely acknowledged safety signals like nightmares and dry mouth emerges from our analysis. Given the complex nature of some ADRs, like nightmares and worsening of depression, further studies are required to better understand their implications for patient safety.

## Figures and Tables

**Figure 1 pharmaceuticals-17-00342-f001:**
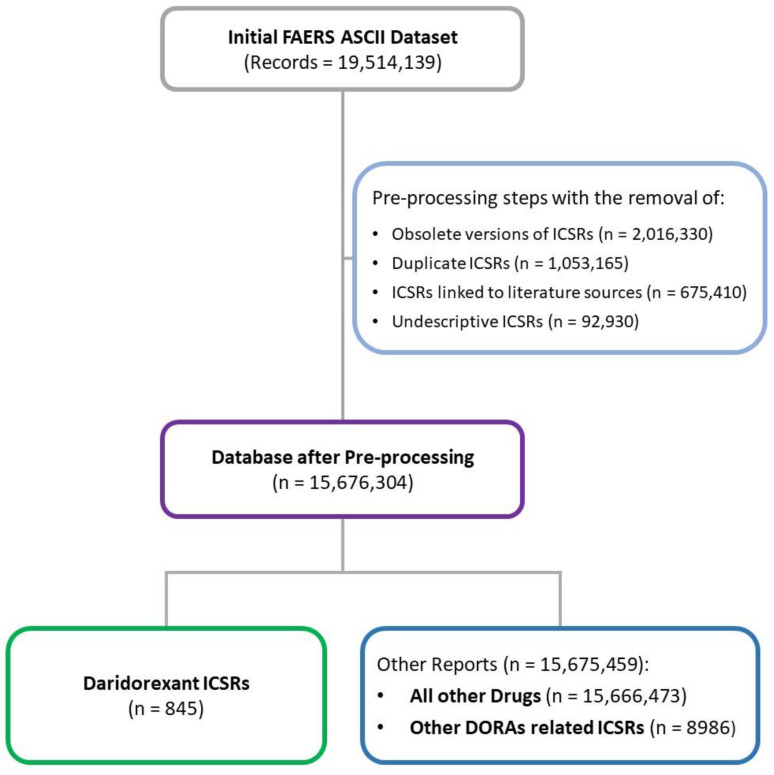
Database preprocessing and individual case safety report (ICSR) selection. ASCII: American standard code for information interchange; DORAs: dual orexin receptor antagonists; ICSRs: individual case safety reports; FAERS: Food and Drug Administration Adverse Event Reporting System.

**Table 1 pharmaceuticals-17-00342-t001:** Demographic characteristics in daridorexant and other dual orexin receptor antagonists related Individual Case Safety Reports.

Characteristic	Daridorexant-Related ICSRs (*n* = 845)	Other DORAs Related ICSRs (*n* = 8986)
Sex, *n*° (%)		
Women	527 (62.4)	4914 (54.7)
Men	275 (32.5)	3012 (33.5)
Not Available	43 (5.9)	1060 (11.8)
Age Group, *n*° (%)		
0–17	-	25 (0.3)
18–34	16 (1.9)	288 (3.2)
35–49	45 (5.3)	591 (6.6)
50–64	101 (12)	1065 (11.9)
65–79	77 (9.1)	1254 (14.0)
80+	22 (2.6)	675 (7.5)
Not available	584 (69.1)	5088 (56.6)
Reporter type, *n*° (%)		
Consumer	712 (84.3)	3426 (38.1)
Healthcare professional	131 (15.5)	5549 (61.8)
Not specified	2 (0.2)	11 (0.1)
Seriousness, *n*° (%)		
Not serious	805 (95.3)	7427 (82.7)
Serious	40 (4.7)	1559 (17.4)
Seriousness Criteria, *n*° (%)		
Death	9 (1.1)	159 (1.8)
Disability	-	73 (0.8)
Hospitalization	3 (0.4)	443 (4.9)
Life-Threatening	1 (0.1)	85 (0.9)
Other Serious Important Medical Event	26 (3.1)	797 (8.9)
Required Intervention	1 (0.1)	2 (0.0)
Not serious	805 (95.3)	7427 (82.7)

Dari: daridorexant; ICSRs: individual case safety reports.

**Table 2 pharmaceuticals-17-00342-t002:** Disproportionality analysis with daridorexant-related adverse drug reactions regrouped by High Level Group terms.

HLGT	ICSRs ^a^	Dari vs. RG1 ROR [95%CI]	Dari vs. RG2 ROR [95%CI]
Sleep disorders and disturbances	326	26.42 [23.01, 30.35]	1.84 [1.59, 2.13]
Sleep disturbances (incl subtypes)	32	20.63 [14.49, 29.37]	0.91 [0.63, 1.31]
Disturbances in thinking and perception	44	7.15 [5.28, 9.68]	1.13 [0.82, 1.55]
Therapeutic and nontherapeutic effects (excl toxicity)	311	4.74 [4.12, 5.46]	1.05 [0.91, 1.21]
Changes in physical activity	10	4.36 [2.34, 8.14]	1.88 [0.95, 3.69]
Product quality, supply, distribution, manufacturing, and quality system issues	69	3.98 [3.11, 5.08]	4.12 [3.1, 5.47]
Salivary gland conditions	14	3.38 [1.99, 5.73]	2.21 [1.24, 3.94]
Headaches	69	2.34 [1.83, 3]	1.66 [1.27, 2.15]
Anxiety disorders and symptoms	40	1.79 [1.31, 2.46]	1.01 [0.73, 1.41]
Depressed mood disorders and disturbances	25	1.73 [1.16, 2.57]	1.98 [1.29, 3.06]
Suicidal and self-injurious behaviors NEC	16	1.63 [0.99, 2.67]	0.96 [0.57, 1.61]
Neurological disorders NEC	114	1.61 [1.32, 1.96]	1 [0.81, 1.23]
Movement disorders (incl parkinsonism)	26	1.48 [0.96, 2.18]	1.16 [0.77, 1.75]
Cardiac disorders, signs and symptoms NEC	16	1.46 [0.89, 2.39]	1.56 [0.92, 2.64]
Overdoses and underdoses NEC	23	1.45 [0.96, 2.19]	0.85 [0.55, 1.3]
Allergic conditions	30	1.43 [0.94, 2.07]	6.71 [4.24, 10.63]
General system disorders NEC	194	1.31 [1.12, 1.54]	2.11 [1.77, 2.5]
Lifestyle issues	10	1.2 [0.64, 2.23]	3.25 [1.6, 6.62]
Deliria (incl confusion)	11	1.19 [0.66, 2.16]	0.6 [0.33, 1.11]
Medication errors and other product use errors and issues	102	1.18 [0.96, 1.45]	1.21 [0.97, 1.51]
Skin appendage conditions	21	0.94 [0.61, 1.44]	2.49 [1.54, 4.03]
Mood disorders and disturbances NEC	11	0.7 [0.39, 1.27]	0.98 [0.53, 1.83]
Muscle disorders	14	0.56 [0.33, 0.95]	0.95 [0.55, 1.64]
Cardiac and vascular investigations (excl enzyme tests)	11	0.54 [0.3, 0.98]	1.23 [0.66, 2.31]
Gastrointestinal signs and symptoms	41	0.54 [0.39, 0.73]	1.25 [0.9, 1.74]
Gastrointestinal motility and defaecation conditions	21	0.52 [0.34, 0.81]	1.65 [1.03, 2.62]
Epidermal and dermal conditions	34	0.49 [0.35, 0.69]	2.03 [1.4, 2.95]
Respiratory disorders NEC	22	0.45 [0.29, 0.69]	1.33 [0.85, 2.08]

Dari: daridorexant; ICSR: individual case safety report; HLGT: High Level Group term; NEC: not elsewhere classified; RG1: Reference Group 1 (individual case safety reports regarding all other drugs in the database); RG2: Reference Group 2 (individual case safety reports regarding other dual orexin receptor antagonists). ^a^ HLGTs with less than 10 associated ICSRs were excluded from the table. Results ordered by reporting odds ratios versus RG1 values, in decreasing order.

**Table 3 pharmaceuticals-17-00342-t003:** Adverse drug reactions belonging to the High Level Group term “sleep disorders and disturbances” classified by Preferred Terms.

PT	ICSRs ^a^	Dari vs. RG1 ROR [95%CI]	Dari vs. RG2 ROR [95%CI]	Exp ^b^
Nightmare	146	113.74 [95.13, 136]	2.35 [1.93, 2.85]	Yes
Insomnia	80	7.41 [5.88, 9.33]	1.94 [1.51, 2.49]	No
Abnormal dreams	64	52.79 [40.9, 68.13]	1.16 [0.89, 1.52]	Yes
Somnambulism	18	56.94 [35.67, 90.9]	1.51 [0.91, 2.48]	Yes
Middle insomnia	15	19.82 [11.89, 33.03]	0.98 [0.58, 1.68]	No
Poor quality sleep	10	11.51 [6.17, 21.48]	0.55 [0.29, 1.04]	No
Sleep terror	9	61.64 [31.93, 119.02]	1.85 [0.91, 3.77]	No
Sleep-related eating disorder	4	113.13 [42.23, 303.02]	2.67 [0.89, 7.99]	Yes
Initial insomnia	4	9.91 [3.71, 26.46]	0.6 [0.22, 1.64]	No
Hypnagogic hallucination	3	267.22 [85.32, 836.95]	4 [1.06, 15.1]	Yes

Dari: daridorexant; ICSR: individual case safety report; Exp: expected; PT: Preferred Term; RG1: Reference Group 1 (ICSRs regarding all other drugs in the database); RG2: Reference Group 2 (ICSRs regarding other dual orexin receptor antagonists). ^a^ PTs with less than 3 associated ICSRs were excluded from the table. ^b^ ADR expectedness evaluated in accordance with US Food and Drug Administration (FDA) and European Medicine Agency Summary of Product Characteristics (SmPCs). Results ordered by ICSR frequency in decreasing order.

**Table 4 pharmaceuticals-17-00342-t004:** Adverse drug reactions belonging to the High Level Group term “general system disorders NEC” classified by Preferred Terms.

PT	ICSRs ^a^	Dari vs. RG1 ROR [95%CI]	Dari vs. RG2 ROR [95%CI]	Exp ^b^
Feeling abnormal	64	6.19 [4.8, 7.99]	1.75 [1.33, 2.29]	No
Fatigue	45	1.41 [1.05, 1.91]	2.28 [1.64, 3.17]	Yes
Hangover	20	127.92 [81.98, 199.62]	3.38 [2.04, 5.61]	Yes
Illness	17	7.19 [4.44, 11.62]	18.43 [8.41, 40.38]	No
Malaise	12	0.61 [0.34, 1.07]	1.07 [0.59, 1.95]	No
Feeling jittery	6	6.89 [3.08, 15.37]	1.25 [0.54, 2.93]	No
Asthenia	6	0.37 [0.17, 0.83]	1.16 [0.5, 2.71]	No
Energy increased	5	18.7 [7.76, 45.06]	6.68 [2.18, 20.47]	No
Crying	5	2.95 [1.22, 7.1]	2.67 [0.99, 7.13]	No
Concomitant disease aggravated	4	15.43 [5.78, 41.23]	-	No
Sluggishness	4	9.02 [3.38, 24.09]	3.05 [0.99, 9.28]	Yes
Chest pain	4	0.48 [0.18, 1.28]	1.29 [0.46, 3.65]	No
Pain	4	0.14 [0.05, 0.38]	0.82 [0.29, 2.26]	No
Screaming	3	10.74 [3.46, 33.39]	2.66 [0.75, 9.46]	No
Discomfort	3	1.16 [0.37, 3.6]	2.91 [0.81, 10.44]	No
Swelling face	3	1.06 [0.34, 3.3]	2.91 [0.81, 10.44]	No
Condition aggravated	3	0.27 [0.09, 0.83]	1.88 [0.55, 6.43]	No

Dari: daridorexant; ICSR: individual case safety report; Exp: expected; PT: Preferred Term; RG1: Reference Group 1 (ICSRs regarding all other drugs in the database); RG2: Reference Group 2 (ICSRs regarding other dual orexin receptor antagonists). ^a^ PTs with less than 3 associated ICSRs were excluded from the table. ^b^ ADR expectedness evaluated in accordance with the USA Food and Drug Administration (FDA) and European Medicine Agency Summary of Product Characteristics (SmPCs). Results ordered by ICSR frequency in decreasing order.

## Data Availability

This study was entirely based on publicly anonymized data made available by the Food and Drug Administration. The raw data can be downloaded at the following link: https://fis.fda.gov/extensions/FPD-QDE-FAERS/FPD-QDE-FAERS.html (accessed on 15 May 2023).

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
