# Peer review of "Safety of Dual Orexin Receptor Antagonist Daridorexant: A Disproportionality Analysis of Publicly Available FAERS Data"

_pharmaceuticals, 2024, doi:10.3390/ph17030342_

Round 1
Reviewer 1 Report
Comments and Suggestions for Authors
I read with interest the paper titled "Safety of Dual Orexin Receptor Antagonist Daridorexant: A Disproportionality Analysis of Publicly Available FAERS Data"
The paper provide a really good background on the real world data of the drug, since in Europe, almost no data is available.
Page 8 - consider to simplify and not repeat too much the data from the table in the text. It's hard to read and the information it's already in the table. If highlighted in the text, keep the description simple - too much numbers in the last two paragraphs.
Please refer that table 2 is ordered by the ROR of RG1.
Please refer that table 3 and 4 is ordered by the frequency of ICSR.
Please discuss further the pattern found for allergic conditions, skin conditions, epidermal conditions (seems to me higher in RG2 than RG1, with a solid difference). Any explanation on the mechanism of Daridorexant vs other DORAs? Why such increased ROR (compared with other DORAs and the whole database)? Are them expected? When use Dari, shoudl we expect skin/dermal/allergic reaction with clinical significance in the skin (like rash, erithrema...)?
Author Response
Reviewer 1
I read with interest the paper titled "Safety of Dual Orexin Receptor Antagonist Daridorexant: A Disproportionality Analysis of Publicly Available FAERS Data". The paper provides a really good background on the real-world data of the drug since, in Europe, almost no data is available.
Thank you for your valuable input which has significantly contributed to the enhancement of our manuscript's quality and the improvement of its readability.
- Page 8 - consider to simplify and not repeat too much the data from the table in the text. It's hard to read and the information it's already in the table. If highlighted in the text, keep the description simple - too much numbers in the last two paragraphs.
Thank you for this comment. We apologize for the lack of readability of this section. We have condensed the descriptions of the results in the last two paragraphs of page 8, line 222 to 233:
“As far as PTs belonging to the HLGT depressed mood disorders and disturbances those were: depression (n = 22; dari vs. RG1: ROR = 2.13; 95%CI[1.39, 3.25] and depressed mood (n = 3; dari vs. RG1: 1.36; [0.44, 4.24]); see Table S3 for more information. Both ADRs are acknowledged in European or FDA SmPC. Frequently co-reported PTs with depression were drug ineffective (n = 7; 31.8%), suicidal ideation (n= 5; 22.7%), and insomnia (n = 4; 18.2%). A positive dechallenge was observed in 45.5% of depression cases (n = 10) and a negative one in 9.1% (n = 2). No rechallenge data were available. TTO data was available in 6 ICSRs (27.3%), with a median value of 5 days IQR[1-11].
Considering the HLGT General system disorders NEC, disproportional PTs were feeling abnormal (n = 64 dari vs. RG1: ROR = 6.19; 95%CI[4.8, 7.99]), followed by fatigue (n= 45 dari vs. RG1: 1.41; [1.05, 1.91]), hangover (n = 20; dari vs. RG1: 127.92; [81.98, 199.62), and energy increased (n = 5; dari vs. RG1: 18.7; [7.76, 45.06]); see Table 4 for further details.”
- Please refer that table 2 is ordered by the ROR of RG1.
Thank you, clarified by adding “Results ordered by Reporting Odds Ratios versus RG1 values, in decreasing order” to the footnote of Table 2.
- Please refer that table 3 and 4 is ordered by the frequency of ICSR.
Thank you, clarified by adding “Results ordered by ICSR frequency in decreasing order” to the footnotes of Tables 3 and 4.
- Please discuss further the pattern found for allergic conditions, skin conditions, epidermal conditions (seems to me higher in RG2 than RG1, with a solid difference). Any explanation on the mechanism of Daridorexant vs other DORAs? Why such increased ROR (compared with other DORAs and the whole database)? Are them expected? When using Dari, should we expect skin/dermal/allergic reaction with clinical significance in the skin (like rash, erythema...)?
Thank you for this comment. In our study, we proposed a two-step process; the first aimed to select only HLGTs with RORs significant both if compared with the rest of the database (RG1) and to other DORAs (RG2). Thereafter, only PTs related to these HLGTs with significant RORs were evaluated in detail. The examples provided by the reviewer did not formally satisfy the first step. Thus, they did not appear in the results section. However, the HLGT allergic condition is effectively almost significant and, following your suggestion, we altered the discussion section of the revised version of the manuscript to include a brief discussion concerning it (page 12, lines 382-387), as follows:
“An almost significant ROR comparing dari to RG1 was observed for the HLGT “allergic conditions”. Hypersensitivity was the only disproportional event (data not shown). In three cases, this was co-reported with rash, while in the other two cases with pruritus and erythema. This common pattern suggests further investigations to properly assess the role of dari in allergic or cutaneous reactions and their clinical impact.”
Reviewer 2 Report
Comments and Suggestions for Authors
This manuscript presents the findings of a study titled "Safety of Dual Orexin Receptor Antagonist Daridorexant: A Disproportionality Analysis of Publicly Available FAERS Data." The topic is pertinent, and the study's design and procedures are clearly articulated. The article effectively communicates its aim. However, improvements are needed in terms of bibliographic references, clarity in presenting certain results, and the spelling of abbreviations used.
Abstract: The authors refer to Daridorexant as "Dari." To avoid confusion for readers, it's recommended that the first mention of Daridorexant include "(Dari)" in parentheses. This practice should be extended throughout the article. Additionally, the authors should spell out the acronym "FDA" upon first use.
Introduction:
Line 29: "The Fifth edition of the Diagnostic and Statistical Manual of Mental Disorders defines insomnia as a condition characterized by a predominant dissatisfaction with sleep quantity or quality." The authors must provide a proper reference.
Line 68: "Daridorexant is the latest DORA introduced on the market and the only one to be authorized for the treatment of insomnia by the European Medicines Agency (EMA)." Adequate referencing is required.
Results:
Line 104: "Regarding demographic characteristics, daridorexant-related ICSRs showed a woman-to-man patient ratio of 1.9 (n = 533; 62.3% vs. n = 279; 32.6%)." These values conflict with those in Table 1. The authors should clarify this discrepancy.
Line 115: "Among serious ADRs, those categorized as 'other serious important medical events' (n = 26; 65.0%) were the most represented, followed by ICSRs in which the death of the patient was reported (n = 9; 22.5%) and ICSRs that described an ADR causing or prolonging a hospitalization (n = 3; 7.5%)." The authors should ensure consistency in calculating percentages to avoid confusion. The percentages in the table1 are different.
Table 1: The statement "Not available" It doesn't make sense to say this if this refers to non-serious ICSRs. Additionally, the acronym "DORAs" is missing in the table header.
Table 2: "HRespiratory disorders NEC" should be "Respiratory disorders NEC." The authors should provide explanations for "RG1" and "RG2" in the footnotes to enhance clarity.
Authors should refrain from using abbreviations in table titles to prevent misinterpretations by readers who have not read the full article.
Material and Methods:
Adequate references are lacking in these sections:
Line 413-“A retrospective observational study was conducted by using ICSRs collected in the FAERS database. FAERS is a widely utilized publicly accessible database comprising over 19 million ICSRs from patients, healthcare professionals, and pharmaceutical companies across the United States, Europe, and Asia. These ICSRs, aside from unique identification numbers, can include information, dates of ICSR submission and ADR occurrences, reporting country, primary source qualifications, patient characteristics (such as gender, age, and weight), suspected and concurrent medications along with their prescribed uses, ADR descriptions, and their seriousness.”
Line 437 ” All the observed ADRs have been categorized in accordance with the MedDRA® . MedDRA® provides a multiaxial hierarchical classification articulated in 5 levels. The terms range from the more specific Lowest Level Term to the more encompassing System Organ Class. For this analysis ADRs were at first regrouped in HLGTs and then examined at the PT level.”
Overall, attention to these suggestions will enhance the clarity and quality of the manuscript.
Comments on the Quality of English LanguageModerate editing of English language required
Author Response
Reviewer 2
This manuscript presents the findings of a study titled "Safety of Dual Orexin Receptor Antagonist Daridorexant: A Disproportionality Analysis of Publicly Available FAERS Data." The topic is pertinent, and the study's design and procedures are clearly articulated. The article effectively communicates its aim. However, improvements are needed in terms of bibliographic references, clarity in presenting certain results, and the spelling of abbreviations used.
Thanks for all your insightful comments that helped us improve the manuscript's quality, bibliographical support, and clarity.
- Abstract: The authors refer to Daridorexant as "Dari." To avoid confusion for readers, it's recommended that the first mention of Daridorexant include "(Dari)" in parentheses. This practice should be extended throughout the article. Additionally, the authors should spell out the acronym "FDA" upon first use.
Thank you, the acronyms have been specified and consolidated throughout the entire manuscript.
Introduction:
- Line 29: "The Fifth edition of the Diagnostic and Statistical Manual of Mental Disorders defines insomnia as a condition characterized by a predominant dissatisfaction with sleep quantity or quality." The authors must provide a proper reference.
Thank you, the following reference was added:
American Psychiatric Association Diagnostic and Statistical Manual of Mental Disorders; American Psychiatric Association, 2013; ISBN 0-89042-555-8.
- Line 68: "Daridorexant is the latest DORA introduced on the market and the only one to be authorized for the treatment of insomnia by the European Medicines Agency (EMA)." Adequate referencing is required.
Thank you, reference added:
European Commission Union Register of Medicinal Products for Human Use Available online: https://ec.europa.eu/health/documents/community-register/html/index_en.htm (accessed on 1 February 2024).
Results:
- Line 104: "Regarding demographic characteristics, daridorexant-related ICSRs showed a woman-to-man patient ratio of 1.9 (n = 533; 62.3% vs. n = 279; 32.6%)." These values conflict with those in Table 1. The authors should clarify this discrepancy.
Thank you for your comment. The discrepancy was generated from a typing error deriving from a previous version of the manuscript that went unnoticed. The mistake has been corrected in page 3, lines 104-105, as follows:
“Regarding demographical characteristics, dari-related ICSRs showed a woman-to-man patient ratio of 1.9 (n = 527; 62.4% vs. n = 275; 32.5%)”
- Line 115: "Among serious ADRs, those categorized as 'other serious important medical events' (n = 26; 65.0%) were the most represented, followed by ICSRs in which the death of the patient was reported (n = 9; 22.5%) and ICSRs that described an ADR causing or prolonging a hospitalization (n = 3; 7.5%)." The authors should ensure consistency in calculating percentages to avoid confusion. The percentages in the table 1 are different.
Thank you for your comment. We apologize for the lack of clarity. The values reported in the text were relative to percentages obtained considering only “serious ADR” cases. As suggested, the percentage values have been aligned with those in Table 1 to improve readability in pages 3-4, lines 114-118:
“Among serious ADRs, those categorized as “other serious important medical events” (n = 26; 3.1%) were the most represented, followed by ICSRs in which the death of the patient was reported (n = 9; 1.1%) and ICSRs that described an ADR causing or prolonging a hospitalization (n = 3; 0.4%). See Table 1 for further details.”
- Table 1: The statement "Not available" It doesn't make sense to say this if this refers to non-serious ICSRs. Additionally, the acronym "DORAs" is missing in the table header.
Thank you, the statement has been changed to not serious in Table 1.
- Table 2: "HRespiratory disorders NEC" should be "Respiratory disorders NEC." The authors should provide explanations for "RG1" and "RG2" in the footnotes to enhance clarity.
Thank you, the typing error has been corrected, and the following descriptions of RG1 and RG2:
“RG1: Reference Group 1 (Individual Case Safety Reports regarding all other drugs in the database); RG2: Reference Group 2 (Individual Case Safety Reports regarding other dual orexin receptor antagonists)” have been added to the footnotes of Tables 2, 3 and 4.
- Authors should refrain from using abbreviations in table titles to prevent misinterpretations by readers who have not read the full article.
Thank you, following your suggestion, table titles have been modified, accordingly.
Material and Methods:
Adequate references are lacking in these sections:
- Line 413- “A retrospective observational study was conducted using ICSRs collected in the FAERS database. FAERS is a widely utilized publicly accessible database comprising over 19 million ICSRs from patients, healthcare professionals, and pharmaceutical companies across the United States, Europe, and Asia. These ICSRs, aside from unique identification numbers, can include information, dates of ICSR submission and ADR occurrences, reporting country, primary source qualifications, patient characteristics (such as gender, age, and weight), suspected and concurrent medications along with their prescribed uses, ADR descriptions, and their seriousness.”
Thank you, the following references were added:
Khaleel, M.A.; Khan, A.H.; Ghadzi, S.M.S.; Adnan, A.S.; Abdallah, Q.M. A Standardized Dataset of a Spontaneous Adverse Event Reporting System. Healthcare 2022, 10, 420, doi:10.3390/healthcare10030420.
Food and Drug Administration Questions and Answers on FDA’s Adverse Event Reporting System (FAERS) Available online: https://www.fda.gov/drugs/surveillance/questions-and-answers-fdas-adverse-event-reporting-system-faers (accessed on 30 November 2023).
- Line 437 ”All the observed ADRs have been categorized in accordance with the MedDRA®. MedDRA® provides a multiaxial hierarchical classification articulated in 5 levels. The terms range from the specific Lowest Level Term to the more encompassing System Organ Class. For this analysis ADRs were at first regrouped in HLGTs and then examined at the PT level.”
Thank you, the following reference was added:
Brown, E.G.; Wood, L.; Wood, S. The Medical Dictionary for Regulatory Activities (MedDRA). Drug Saf 1999, 20, 109–117, doi:10.2165/00002018-199920020-00002.
Round 2
Reviewer 2 Report
Comments and Suggestions for Authors
The authors clearly improved the manuscript, therefore, in my opinion, it is now susceptible for publication
Comments on the Quality of English LanguageMinor editing of English language required